# A Novel Approach for Simulation and Optimization of Rubber Vulcanization

**DOI:** 10.3390/polym15071750

**Published:** 2023-03-31

**Authors:** Jelena Lubura, Predrag Kojić, Jelena Pavličević, Bojana Ikonić, Dario Balaban, Oskar Bera

**Affiliations:** Faculty of Technology Novi Sad, University of Novi Sad, Bulevar cara Lazara 1, 21 000 Novi Sad, Serbia

**Keywords:** simulation, vulcanization, molding, rubber, computer modeling

## Abstract

The kinetic model, encompassing the curing and reversion phenomena of the NR/SBR rubber vulcanization process, was developed by means of the finite element method simultaneously with heat transfer equations, including heat generation due to curing reactions. The vulcanization simulation was conducted for three spheres of different diameters (1, 5 and 10 cm) and two rubber wheels, one of which was a commercial product of the rubber industry. The proposed advanced simulation model, based on products’ two-dimensional axisymmetry, includes cooling after vulcanization, during which the crosslinking reactions continue to take place as a result of the products’ heated interiors. As a criterion for removing the product from the mold, an average vulcanization degree of 0.9 was set, whereby, during cooling, the vulcanization degree increases, due to crosslinking reactions. Based on the minimal difference between the maximal and minimal vulcanization degrees, which did not exceed a value of 0.0142, the optimal process parameters for each product were determined, achieving homogeneity and obtaining high-quality rubber products, while simultaneously ensuring a more efficient vulcanization process and enhanced cost effectiveness for the rubber industry.

## 1. Introduction

Rubber products are widely used in a broad range of applications when their highly deformable characteristics are required. Natural rubber and styrene–butadiene rubber blend have been extensively used to optimize the compound properties having the required mechanical–dynamical properties of the final products [1,2]. Soft raw rubber must be cured in a mold in a process named vulcanization. A compound containing uncured rubber and curing agents fills the mold cavity, and are heated to the temperature where crosslinking reactions take place [3,4]. The vulcanization is a fundamental step in rubber production and plays an important role in achieving a defined shape, and the physical and mechanical properties of the final product. The vulcanization process and molding require heating for predefined periods of time, leading to energy consumption. Optimization of the parameters of the energy-demanding vulcanization process is of crucial importance, in order to achieve economical process management and to obtain high quality rubber products. The low thermal diffusivity of the rubber mixture leads to non-uniform and time-varying temperature fields inside the product. Traditionally, the experimental method used to determine a product’s thermal history involved measuring the temperature at different points within the product, calculating the curing degree using the kinetic equation, and, finally, determining the necessary vulcanization time. This experimental method for obtaining the curing degree was time-consuming and expensive, since it was necessary to damage the rubber product in order to measure the internal temperature. Nowadays, the preferred technique in the modern rubber industry for assessment of the thermal history inside the product makes use of heat transfer equations with an appropriate numerical method [5,6]. Approaches based on computer vulcanization simulation allow determination of the optimal process parameters, such as temperature and time, without the need for expensive and time-consuming experiments. By optimizing the process, a higher vulcanization degree can be achieved in a short time and at a lower operating temperature, leading to higher process profitability, reduced energy consumption, and quality products that meet the high rubber market standards [7].

The simulation of the vulcanization process requires the application of an appropriate kinetic equation that describes the process with minimal deviation. Additionally, it is important that the chosen equation is consistent with vulcanization theory and simple to solve. Modeling and optimization of rubber product vulcanization are complex due to the induction and reversion phenomena. Numerous different approaches can be found in the literature for rubber kinetics studies [8,9,10,11,12,13,14,15]. Empirical, mechanistic and semi-mechanistic models represent novel kinetic models that take into account reversion phenomena. Milani and Milani [16] presented an empirical model, based on parabolic and hyperbolic equations, but the model’s main disadvantage is the lack of the parameters’ physical meaning. Ding and Leonov proposed a mechanistic model [17] including reversion and enabling complete vulcanization modeling. However, it consists of complex reactions with more than six rate constants, which can lead to it being a complex model to solve. Simplified reaction mechanisms, enabling a reduced number of kinetic parameters, were proposed by Han [18] and Milani [19,20] in semi-mechanistic models. Temperature is one of the key parameters during vulcanization, since it significantly affects the crosslinking density and the type of crosslinks formed. Furthermore, structure, mechanical properties and thermal stability of the final rubber product significantly depend on the process parameters, and, therefore, the simulation and optimization of the vulcanization process are of crucial importance.

Three approaches can be used for vulcanization modeling and simulation: the analytical method, the finite difference method (FDM) and the finite element method (FEM). Analytical methods are not commonly used to solve the heat transfer and vulcanization kinetics equations, due to the complex shape of most rubber products [5]. The FDM is the simplest numerical method used to solve heat transfer partial differential equations and to estimate the temperature profile within the product. Nevertheless, there are disadvantages because the FDM faces difficulties when modeling complex geometries [21]. The FEM enables the vulcanization optimization of rubber products under complex conditions and, hence, it is extensively used to solve heat transfer equations [22,23,24,25,26,27,28,29,30,31,32,33,34]. With advanced vulcanization simulation, specific values of various reaction parameters and crosslinking structures can be determined at any time for any point in the vulcanization field, enabling effective vulcanization prediction.

In order to model and simulate the vulcanization of rubber products of different shapes and dimensions in a three-dimensional geometry, the curing and reversion differential equations, as well as the heat transfer equations for the corresponding product geometry, were solved using the finite element method. An investigation of the effect of product size on optimal vulcanization process parameters was performed on three spheres of different selected diameters, as well as on two rubber wheels, one of which was commercially produced in the rubber industry. The optimal vulcanization time and temperature of all tested products were determined using the proposed model and a new optimization procedure that includes curing, reversion and cooling after removing the product from the mold. The developed optimization procedure takes into account cooling after removing the product from the mold, which addresses a critical factor that has been overlooked in previous optimization methods that typically assume the product cools instantaneously after removal from the mold. The novel optimization procedure can reduce the energy consumption required for the vulcanization by optimizing the process parameters.

## 2. Materials and Methods

### 2.1. Materials

The properties (specific heat, density, thermal conductivity coefficient and thermal diffusivity coefficient) of a commercially available mixture of natural and styrene–butadiene rubber, based on a conventional vulcanization system, having sulfur as the curing agent, N-cyclohexyl-2-benzothiazyl-sulfenamide (CBS) as the accelerator, zinc oxide and stearic acid as the curing activators, and 40 phr of carbon black as the filler, were assessed in previous research and are presented in Table 1. The density, thermal conductivity coefficient, and thermal diffusivity coefficient were treated as constants, based on a review of relevant literature [5,35]. This decision did not significantly affect the simulation results, as the standard deviation for the constant values of the thermal conductivity coefficient and the thermal diffusivity coefficient were only 0.0074 W m${}^{-1}$ K${}^{-1}$ and 0.0036 mm${}^{2}$ s${}^{-1}$, respectively. Attempting to incorporate an appropriate temperature and vulcanization degree dependence for these parameters resulted in a divergence in certain simulations. Therefore, the decision to use constant values was based on low standard deviation and avoidance of divergence.

### 2.2. Simulation

Numerical simulation of the rubber mixture vulcanization process was performed using COMSOL Multiphysics software, which is extensively applied in various engineering fields, since it provides detailed information on the studied phenomenon for each point of an investigated process [36,37]. COMSOL Multiphysics software is based on the finite element method solving the governing equations and contains an extensive set of functions for analyzing and displaying simulation solutions. In this work, the investigated physical interface was heat transfer in solid materials. As vulcanization depends primarily on the process temperature and time, the simulation was time-dependent.

The first step in creating a simulation model is selecting geometry for the model component, that can be three-dimensional, two-dimensional axisymmetric, two-dimensional, one-dimensional axisymmetric or dimensionless. In this work, two-dimensional axisymmetry was used, whereby all the examined products were symmetrical in cross-section. The choice of this geometry, instead of three-dimensional geometry, was based on the fact it provides fast and precise calculation. The vulcanization simulation was investigated for spheres of different diameters (*D* = 1, 5, 10 cm) and for two rubber wheels with different bases, named Product 1 and Product 2 (cross-sections shown in Figure 1).

The necessary parameters for the simulation were obtained from [38], achieved by fitting. The values of the vulcanization parameters are ($E_{au}$, $E_{ar}$, $A_{u}$, $A_{r}$, $K_{n}$, $N_{n}$, $K_{x}$, $N_{x}$) shown in Table 1. Fitting of the vulcanization curve was performed by the least squares method, described in detail in a prior study [38]. Additionally, the varied parameter was the mold temperature ($T_{M}$), that had a different value for each individual simulation and was then kept constant at that value during each vulcanization simulation. In order to save energy, during the vulcanization simulation, cooling of the rubber product after removal from the mold was included. Thus, the vulcanization did not only take place in the molds; the curing reactions inside the product continued during cooling, due to high internal temperatures. The constraint, in order to achieve stability of the rubber product during removal from the mold, required reaching an average vulcanization degree of 0.9. This approach ensured that sufficient product volume was above the vulcanization degree threshold of 0.9. The ambient temperature ($T_{amb}$) was defined by Equation (Equation 7), according to the average vulcanization degree of 0.9, in a way that ensured a product with a defined shape. During vulcanization, the ambient temperature was equal to the mold temperature, and when the product was removed from the mold, the ambient temperature was equivalent to $T_{0}$ = 25 °C. Equation (Equation 7) allowed the average value of the vulcanization degree, in the entire product volume, to reach 0.9; while, for the farthest point from the heat source, the vulcanization degree was lower than the defined value. During cooling, the curing reactions continued to take place due to the hot inner product part, and the minimal vulcanization degree during cooling reached values above 0.9 if vulcanization was performed at the appropriate process parameters.

The heat transfer coefficient was defined by Equation (Equation 6), according to the same principle. During the vulcanization, the heat transfer coefficient between the steel (mold) and the rubber mixture was 10,000 Wm${}^{-2}$K${}^{-1}$ [31]; while, during cooling, the heat transfer coefficient from the rubber product to the air was 11.4 Wm${}^{-2}$K${}^{-1}$ [39].

Theoretically, the reversion degree values cannot be lower than 0 or higher than 1. The reversion degree was defined by Equation (Equation 12), which allowed the reversion degree value to be higher than 0. Values of $K_{x}$ and $N_{x}$ at vulcanization temperatures do not lead to a reversion degree value higher than the maximal theoretical value. The reaction order was defined by Equation (Equation 11) in a way that ensured the reaction order value was not less than 1. The vulcanization degree was defined according to Equation (Equation 10), where the vulcanization degree represents the difference between curing and reversion.

The variables with dependencies on time, defined as a result of the vulcanization simulation, and obtained by the surface integral in the given domain of the material, were:Minimal vulcanization degree ($\alpha_{min}$);Average vulcanization degree ($\alpha_{a}$);Maximal vulcanization degree ($\alpha_{max}$);Minimal temperature ($T_{min}$);Average temperature ($T_{a}$);Maximal temperature ($T_{max}$);Minimal vulcanization rate ($v_{min}$);Average vulcanization rate ($v_{a}$);Maximal vulcanization rate ($v_{max}$).

As part of the physical interface, the thermal properties of the rubber material ($C_{p}$, ρ, λ, *K*) were defined in accordance with Table 1. The solved governing heat transfer equation was defined by Equation (Equation 1).

As the curing reaction is exothermic, the heat obtained from the curing reactions is generated per unit material volume (*Q*), and defined by Equation (Equation 2), where $q_{v}$ is the vulcanization specific heat ($q_{v}$ = 13 kJ kg${}^{-1}$ [40]). The heat of reversion was neglected due to the slow rate of reversion reactions. As the rubber mixture was placed in closed molds, the only heat source was the mold surface that was at a constant temperature. Heat is transferred from the steel to the rubber compound by convection, heat flux across boundaries (Equation (Equation 4)), and through the rubber sample by conduction (Equations (Equation 2) and (Equation 3)).

Curing and reversion are considered as two parallel phenomena of a complex vulcanization process. The methodology for curing and reversion degree determination was proposed in [38], and the final differential forms defined by Equations (Equation 8) and (Equation 9), respectively. The reversion differential kinetic equation was solved using the initial condition of the reversion degree being zero. However, the zero value curing degree initial condition led to a mathematically indeterminate system. Accordingly, the initial value of the curing degree was set at 0.005, consistent with the theory that rubber is partially crosslinked even before the start of vulcanization [41].

The finite element method involves discrete analysis, and is based on the physical discretization of the selected area, where the basis of the calculation is a part of the area of finite dimensions, or a finite element. The discretization domain has an infinite number of degrees of freedom, and is replaced by a discrete model of interconnected finite elements, with a finite number of degrees of freedom [42]. The discretization approach was used to solve the heat transfer and the vulcanization kinetic equations, since it provides more reliable solutions, compared to other methods. To solve the differential kinetic equations, it is necessary to obtain the dependence of temperature on time in all coordinate points (T(r,z,t)), by solving energy balance equations. In order to perform the simulation, the first derivative of the vulcanization degree is calculated, where the increment depends on the previous state of the vulcanization degree and temperature, for all coordinate points, in the corresponding time. Solutions of the coupled heat transfer and the kinetic model equations are functions of temperature and vulcanization degree on time, for all coordinate points: T(r,z,t) and α(r,z,t), respectively. The parameters necessary for the vulcanization optimization ($T_{min}$, $T_{a}$, $T_{max}$, $\alpha_{min}$, $\alpha_{a}$, $\alpha_{max}$, $v_{min}$, $v_{a}$, $v_{max}$) are integral quantities obtained by the surface integral for each solution over a set of domains in two-dimensional axisymmetry, as a function of time. The simplified overview of the coupled heat transfer and kinetic equations is presented in Table 1 that contains the equations used within the simulation and the defined variables, as well as the solutions of the equations.

One of the key phases for obtaining precise results using FEM is selection of a geometric mesh in which the continuous geometry is decomposed into thousands of elements, or more, in order to properly define the physical shape of the object. Geometric mesh consists of a set of quadrilaterals and triangles that represent a surface or solid geometry. Small mesh elements allow more accurate results but the calculation time might be too long. It is necessary to optimize calculation time, accuracy and convergence of the model, since a choice of mesh can lead to divergence. Therefore, it is important to adequately choose the geometric mesh, as well as the domain in which the geometric mesh is to be used in performing the simulation. In this work, “Physics Controlled Mesh”, with an element size set to “Extra fine”, defined in the software COMSOL Multiphysics, was used, which enabled precise simulation. As the tested products were symmetric along a fixed axis, this type of geometric mesh did not lead to lengthy simulation times when used only in the domain shown in Figure 1. The simulations were carried out using selected physics modules with a controlled tolerance for relative error of 0.01. The absolute tolerance was defined using the “Scaled” Global method with a default tolerance factor of 0.1. In the “Scaled” method, the absolute tolerance is applied to scaled variables, while the “factor tolerance” method makes the absolute tolerance proportional to the relative tolerance. The simulations could be considered precise, in terms of mesh definition and error, as the solver was run using default settings, resulting in a short simulation time of approximately 10 min.

Spheres of diameter 1 and 10 cm have the same domain as a particularly fine geometric mesh, as in the sphere shown in Figure 1. Observing the mesh of Product 2, it can be seen that the geometric mesh was dense on the rounded parts of the product, in order to accurately solve the model equations.

According to the results obtained by optimizing the vulcanization of spheres of different diameters in a previous study, [38], it was assumed that vulcanization of larger products would take a longer time, at lower temperatures. Accordingly, for the vulcanization test, different products required different simulation times. A sphere of diameter 1 cm was simulated in 30 min, Product 2 in 60 min, and Product 1 in 120 min, while a sphere with a diameter of 10 cm was simulated in 180 min, in order to ensure sufficient time for the vulcanization and subsequent cooling of the product to temperatures at which curing reactions can be considered to no longer be taking place, due to a low curing rate. The time step of all simulations was 1 s, and the average simulation time was 10 min.

## 3. Results and Discussion

### 3.1. Simulation of Vulcanization of a Sphere of Different Dimensions

In order to understand the vulcanization simulation, as well as the optimization of process parameters, the simulation procedure is explained using the example of a sphere with a diameter of D= 5 cm, while the results for other shapes and dimensions are presented afterwards. For the vulcanization simulation and cooling of a sphere with diameter D= 5 cm, a time of 120 min was selected, which was long enough to cool the product to a temperature approximately equal to room temperature, ensuring that curing reactions inside the product were no longer taking place, confirmed in Figure 2 and Figure 3. Figure 2 shows the dependence of product temperature on time, where the mold temperature was constant at 150 °C, and cooling was included after removing the product from the vulcanization mold. The values of the minimal, average and maximal temperatures at the moment of removing the product from the mold are shown, and marked as $T_{min,r}$, $T_{a,r}$ and $T_{max,r}$, respectively, where the time of removal from the mold is marked as $t_{r}$ (red line in Figure 2, Figure 3 and Figure 4). As previously explained by Equations (Equation 4) and (Equation 5), the moment when the product is removed from the mold represents the moment when the average vulcanization degree of the product reaches a value of 0.9.

The cold rubber mixture upon contact with the heated mold almost instantly reached the mold temperature of 150 °C (Figure 2), which can be explained by the high heat transfer coefficient between steel and rubber. The mold temperature was kept constant, allowing the surface of the rubber mixture to be exposed to the maximal vulcanization temperature, until the moment when the product was removed. At the beginning of the process, the minimal temperature of the product was approximately constant and gradually started to rise, explained by the time it took for heat to be conducted to the interior of the product. As the moment of removing the product from the mold was defined by the Equation (Equation 7), it could be observed that after 25.2 min, the minimal product temperature almost reached the value of the maximal temperature and its value was 148 °C. After removing the product from the mold, the surface temperature of the rubber product cooled down rapidly, and during cooling, the surface represented the minimal temperature of the product, since it took more time for the interior to cool down to room temperature. Consequently, during cooling, the farthest point from the product surface became the point of maximal temperature. It is important to note that, after removing the product, the maximal temperature begins to decrease, after which a slight peak appears, explained by reaching the maximal temperature inside the product. The previous discussion is presented in Figure 2, wherein the dotted line continues to the solid one with the same gradient, and the solid line continues to the dotted line, i.e., the center heated up longer than the surface cooled down, after the moment of product removal from the mold.

Figure 3 shows the dependence of the vulcanization degree on time, where $\alpha_{min,r}$, $\alpha_{a,r}$ and $\alpha_{max,r}$ represent the minimal, average and maximal vulcanization degrees at the moment of removing the product from the vulcanization molds, respectively.

As the product surface was exposed to a constant temperature (150 °C), the maximal vulcanization degree on the surface rapidly reached the value of 0.9, and the minimal vulcanization degree reached a value of 0.9 only after removal from the mold (Figure 3). Additionally, according to Figure 2, until the moment of removal from the mold, as the surface of the rubber mixture was exposed to the mold temperature, the maximal values of the vulcanization degree could be attributed to the surface of the product, while the minimal values of the vulcanization degree were related to the farthest point from the surface of the product, i.e., to the center of the sphere. At the removal moment, it can be observed that the minimal vulcanization degree was 0.684, which did not meet industry standards. On the other hand, after removing the product from the mold, the maximal vulcanization degree was approximately constant, while the minimal vulcanization degree continued to increase, until reaching a plateau having constant values. This is attributed to the additional curing of the heated interior of the product, which was still at a high temperature, and the curing reactions still taking place. The values of the minimal, average and maximal vulcanization degrees after cooling were 0.934, 0.94 and 0.945, respectively, obtained in order to test whether the product met industrial standards regarding vulcanization degree. The values of the vulcanization degree after cooling the sphere (D= 5 cm) were above 0.93, and the vulcanization degree was high throughout the interior of the sphere, satisfying the criterion set at 0.9.

The dependence of the vulcanization rate (*v*) on time is shown in Figure 4, where $v_{min,r}$, $v_{a,r}$ and $v_{max,r}$ are the minimal, average and maximal vulcanization rates at the moment of removing the products from the vulcanization molds, respectively.

The sudden increase in the maximal vulcanization rate is attributed to curing reactions on the product surface directly exposed to the mold temperature (Figure 4). After reaching the maximal vulcanization rate, it was noticed that there was a gradual decrease in the rate, since time was required for heat conduction to the interior of the product to start the curing reactions. Furthermore, a second peak of the maximal vulcanization rate could be noticed before removing the product from the mold, explained by ensuing curing reactions in the heated inner part of product. After cooling to a temperature below 120 °C, the curing reactions were slow, and the vulcanization rate was approximately equal to zero.

The vulcanization of the product was previously tested at 150 °C, and in order to find the optimal vulcanization process parameters of a sphere D= 5 cm, other vulcanization temperatures were tested. Figure 5 shows the minimal and maximal vulcanization degrees at the moment of removing the rubber product from the mold, respectively, and the criterion for removing the product ($\alpha_{a}=0.9$) is shown by the red line, while the optimal solutions in the Figure 5, Figure 6 and Figure 7 are marked with red symbols.

The value of the maximal vulcanization degree at the rubber product’s removal time was approximately equal at all process temperatures, since the selected temperature range (120 to 180 °C) was high enough for the curing of the tested sphere, up to its maximal value (Figure 5). Product exposure to high temperatures led to rapid achievement of an average vulcanization degree equal to 0.9 and product removal, while the interior of the product was insufficiently cured, indicated at 170 °C, where the minimal vulcanization degree for the product removal time was 0.518, and the removal time was 18.4 min (Figure 5). Nevertheless, if the vulcanization was performed at 180 °C, a higher value of the minimal vulcanization degree was achieved ($\alpha_{min,r}$ = 0.736), due to the greater temperature difference of the surface and the farthest point of the product, and the heat conduction was faster. Temperatures higher than 180 °C were not investigated, since the value of the average vulcanization degree of 0.9 was not reached during vulcanization, and the criterion for removing the product was not achieved. At high temperatures, thermal degradation of the rubber product surface occurred, while the inner part was still non-cured, and when the interior reached a value of 0.9, the surface was already significantly degraded and the average vulcanization degree was lower than the critical value for removing the product from the vulcanizing mold.

A significant part of the curing reactions takes place after vulcanization, during cooling, and the values of the minimal, average and maximal vulcanization degrees after cooling, at different mold temperatures, are shown in Figure 6, marked as $\alpha_{min,c}$, $\alpha_{a,c}$ and $\alpha_{max,c}$, respectively. In order to define the new optimization procedure, the time when the product ws removed from the vulcanization mold is marked with a red line.

For spherical product (D= 5 cm) vulcanized at 180 °C, it was observed that, after cooling, the minimal vulcanization degree was 0.822, which was defined as an insufficiently cured product; although the process time was the shortest (17.9 min), quality products were not obtained (Figure 6). At low mold temperatures (120 and 130 °C), vulcanization degrees higher than 0.9 were achieved, but, nevertheless, the long process time led to higher energy consumption. It can be noticed that at a temperature of 150 °C and $t_{r}=$ 25.2 min, the highest values of the minimal vulcanization degree (0.934) were reached after cooling, and the average and maximal vulcanization degrees were 0.94 and 0.945, respectively, indicating that the entire product volume was highly cured. In order to optimize the process, the difference between the maximal and minimal vulcanization degrees at the time of product removal from the mold ($\Delta \alpha_{r}$) was examined and compared with the values after cooling ($\Delta \alpha_{c}$), as presented in Figure 7.

It can be observed that the difference between the maximal and minimal vulcanization degrees was significantly lower after cooling, compared to the values at the removal time (Figure 7). At low process temperatures, there was a very small difference between the maximal and minimal vulcanization degrees before and after cooling, while, as the temperature increased, the differences were more pronounced. The difference between maximal and minimal vulcanization degrees ($\Delta \alpha_{c}=$ 0.011) was the lowest for the product obtained at 150 °C, in 25.2 min, indicating a homogeneous product after cooling. These process parameters can be defined as an optimization solution for a sphere with a diameter of 5 cm, to obtain a high-quality product in a short time, simultaneously enabling cost efficiency in rubber production. Figure 8 shows the simulation results of a sphere (D= 5 cm), vulcanized at the optimal temperature, at the moment when the product was removed from the mold and after cooling.

At the moment when the product was removed from the mold, the lowest value of the vulcanization degree was observed in the sphere center (α= 0.684), while, on the surface, its value was assessed at 0.943 (Figure 8). After cooling, for the entire sphere volume, the value of the vulcanization degree ranged from 0.934 to 0.945, indicating an homogeneous product. Quality rubber products are fully homogeneous within the entire volume, related to the achievement of a high curing degree throughout the product, i.e., a minimal difference between the maximal and minimal vulcanization degrees. Optimization of process parameters represents finding the minimal difference between the maximal and minimal vulcanization degrees after cooling ($\Delta \alpha_{min,c}$), simultaneously paying attention to time saving and energy efficiency.

The proposed simulation and optimization of the vulcanization process for obtaining high-quality rubber products can be adopted and applied to other products of different shapes and dimensions. The optimization procedure was applied to obtain two additional rubber spheres, with diameters of 1 cm and 10 cm, and the difference between maximal and minimal vulcanization degrees of spheres of all investigated diameters, at different mold temperatures, is shown in Figure 9, where the optimal solutions are marked in red.

By comparing the difference between the maximal and minimal vulcanization degrees after cooling the products obtained at different process temperatures (Figure 9), it was observed that vulcanization could not be performed at the same temperatures to obtain products of different diameters. For products with larger diameters there was a narrow temperature interval at which vulcanization could be conducted. Observing a sphere with a diameter of 10 cm, it can be seen that at lower and higher temperatures than the optimal, the difference between the maximal and minimal vulcanization degrees rapidly increased. At temperatures higher than optimal, there was a sudden increase in the $\Delta \alpha_{min,c}$ value, for spheres of all diameters, which can be explained by thermal degradation of the product surface at high temperatures, while the interior part was not sufficiently cured. At temperatures lower than optimal, the vulcanization required higher energy consumption, due to the longer vulcanization time.

The following equations for calculating the optimal vulcanization temperature and time are proposed [38]:(13)$$T_{opt}=T_{min}+\frac{p}{D}\;{(}^{\circ }C)$$
(14)$$t_{opt}=aD^{b}\;\left(\min\right)$$
where *p*, *a* and *b* are the fitting parameters, and with Equations (Equation 13) and (Equation 14) the constraints and boundary conditions are set. When the diameter of the sphere tends to zero (D→0), then the sphere becomes a point and can be exposed to an infinitely high temperature ($T_{opt}\to \infty$), for an infinitely short time ($t_{opt}=0$). On the other hand, if the sphere diameter is infinitely large (D→∞), then the sphere can be exposed to a minimal temperature ($T_{opt}=T_{min}$) for an infinitely large time ($t_{opt}\to \infty$).

Figure 10 shows the optimal process parameters of spheres for all investigated diameters.

An increase in temperature led to a decrease in vulcanization time (Figure 10). The optimal vulcanization temperature of larger products was significantly lower, compared to the temperature at which small vulcanizates are produced, since the surface of large products degrades with long exposure to heat, while the interior is insufficiently cured. On the contrary, heat is rapidly conducted to the interior of small products, and an adequately crosslinked product obtained. The curves in Figure 10 correspond to Equations (Equation 13) and (Equation 14), and the obtained equation parameters are presented in Table 2. The obtained parameters are specific for the tested rubber mixture and can be applied to the spherical geometry. From Table 2, it can be noticed that the proposed equations adequately describe the dependence of optimal temperature and time on the sphere diameter, confirmed by the numerical methods (high value of $R^{2}$ and low value of *MAPE*).

### 3.2. Simulation of Vulcanization of Rubber Wheels

As the size and shape of the vulcanizate depends on demand in the rubber market, it is necessary to provide homogeneous rubber products of any diameter and shape, achieved by optimization, the process parameters of which can be determined by the proposed optimization procedure that includes cooling after vulcanization. The optimization procedure was applied to two additional shapes, two rubber wheels with different bases. The differences between the maximal and minimal vulcanization degrees for Products 1 and 2, after cooling, are shown in Figure 11, where the optimal process temperatures are marked with red.

It can be observed that high temperatures led to high values of $\Delta \alpha_{c}$, while at lower temperatures the value of $\Delta \alpha_{c}$ was lower, along with a long process time causing high energy consumption (Figure 11). Additionally, for Product 2, it can be observed that the value of $\Delta \alpha_{c}$ at 120 °C was close to the optimal; however, the vulcanization time was 89.3 min, representing ineffective process management, even when quality products are obtained. The vulcanization simulations at optimal process parameters (for Products 1 and 2) are shown in Figure 12 and Figure 13, respectively.

Products 1 and 2 differed in size and shape, and, as previously explained, products with larger diameters require a longer vulcanization time at lower temperatures, compared to products with smaller diameters. This can be observed in the case of rubber wheels (Figure 12 and Figure 13). Nonetheless, Product 2, shown in Figure 13, had a specific shape and it was necessary to specifically consider the process vulcanization parameters, since thin parts on a rubber wheel surface can quickly degrade under high temperatures. Consequently, it can be noted that the optimal process parameters of a rubber wheel were a short exposure time ($t_{v}$ = 5.9 min) to a high temperature 170 °C. Product 1 was larger and the optimal process parameters for obtaining such a vulcanizate were temperature 155 °C and time 18.7 min. The presented results correspond to the range of process parameters in real industrial conditions. The lowest values of the vulcanization degree at the removal time from the mold for Products 1 and 2 were found in the vulcanizate interiors, and their values were 0.779 and 0.781, respectively. After cooling, the lowest vulcanization degree of Product 1 ($\alpha_{c}$ = 0.938) was detected in the interior and on the vulcanizate surface, which could be attributed to surface degradation, due to long exposure to high temperatures, and, as a consequence of Product 1’s dimensions, a longer time was required for the heat to reach the interior. The short vulcanization time of Product 2 at 170 °C did not cause significant thermal degradation, and the vulcanization degree on the surface was 0.942. The lowest values of the vulcanization degree after cooling Product 2 were in the central part of the vulcanizate, and the highest values were between the surface and the central part, as the heat started to be conducted from the surface to the inner part when the product was being removed from the mold. After cooling both products, very homogeneous products were obtained, where the vulcanization degree was reached between 0.938 and 0.944. The proposed vulcanization optimization model, including cooling after removing the product from the mold, can be potentially applied to various shapes of rubber products and can enable the improvement of rubber technology, providing significant energy and time savings in the rubber industry.

## 4. Conclusions

The aim of this work was to develop an advanced model for simulation and optimization of the NR/SBR rubber vulcanization process, including curing and reversion phenomena, as well as cooling after removing the product from the mold, when additional crosslinking reactions take place and high-quality products are obtained. Vulcanization simulations of three spheres with different diameters (*D* = 1, 5 and 10 cm) were performed to examine, in detail, the influence of diameter on vulcanization process parameters. Furthermore, process simulations were performed for two rubber wheels, where one was a product from the rubber industry. The simulations were conducted using the two-dimensional axisymmetry of the products for fast and precise calculations. The moment of removing the product was introduced, defined as the moment when the product reached an average vulcanization degree equal to the value of 0.9. The parameters monitored during the vulcanization simulations were minimal, average and maximal temperature, and vulcanization degree and rate. It was observed that the minimal vulcanization degree, at the moment of removing the product from the mold, was significantly lower than the defined value of 0.9, while it increased during cooling, as a consequence of crosslinking reactions taking place in the hot interior of the rubber product. In order to optimize the vulcanization process, the difference between the maximal and minimal vulcanization degrees after cooling was determined for all products, where the minimal difference between these two parameters represented an homogeneous product. The process parameters for obtaining homogeneous products are the optimal vulcanization temperature and time, taking into account that process conditions should be economical, cost effective and energy efficient. The optimal process parameters of all tested products were determined (Sphere D= 1 cm: $T_{opt}=$ 192 °C, $t_{opt}=$ 1.55 min; Sphere D= 5 cm: $T_{opt}=$ 150 °C, $t_{opt}=$ 25.2 min; Sphere D= 10 cm: $T_{opt}=$ 138 °C, $t_{opt}=$ 73.8 min; Product 1: $T_{opt}=$ 155 °C, $t_{opt}=$ 18.7 min; Product 2: $T_{opt}=$ 170 °C, $t_{opt}=$ 5.9 min), where the differences between the maximal and minimal vulcanization degrees were not greater than 0.0142. The assessed optimization results (vulcanization temperature and time) for the investigated product shapes were successfully tested and confirmed experimentally during rubber production, allowing the application of the proposed optimization approach to obtain vulcanization parameters of various vulcanization systems and rubber products with complex shapes. The proposed parameters resulted in rubber products with the desired properties, while the implementation of optimal parameters in real industrial surroundings enabled reduction of energy consumption.

## Figures and Tables

**Figure 1 polymers-15-01750-f001:**
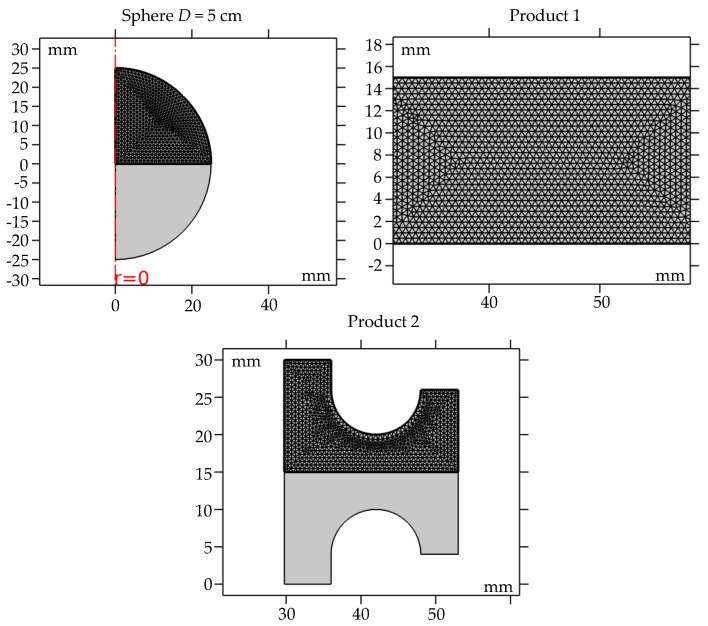
The domain of a particularly fine geometric mesh in selected products.

**Figure 2 polymers-15-01750-f002:**
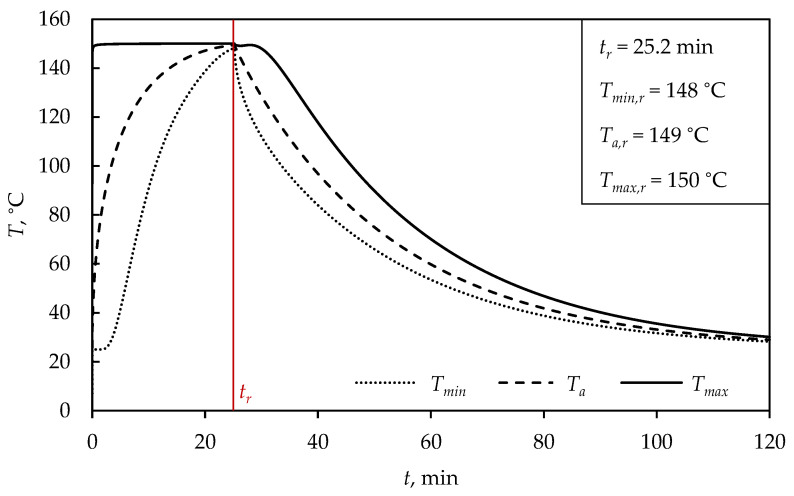
Dependence of product temperature on time for a sphere (D= 5 cm) at $T_{M}=$ 150 °C.

**Figure 3 polymers-15-01750-f003:**
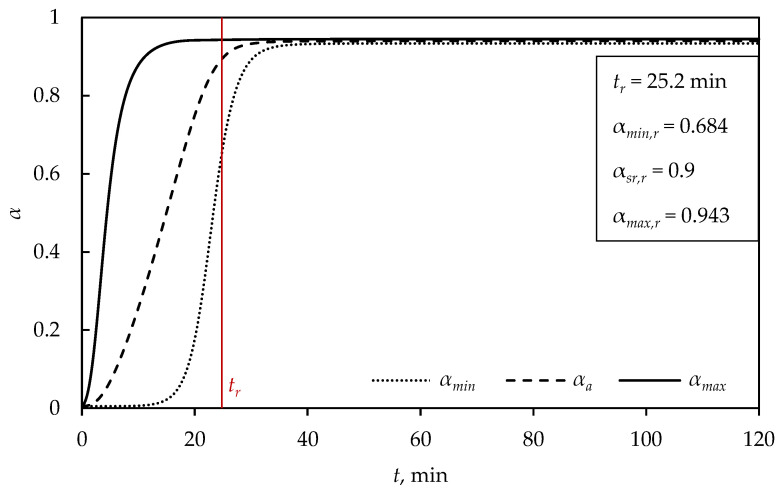
Dependence of the vulcanization degree on time for a sphere (D= 5 cm) at $T_{M}=$ 150 °C.

**Figure 4 polymers-15-01750-f004:**
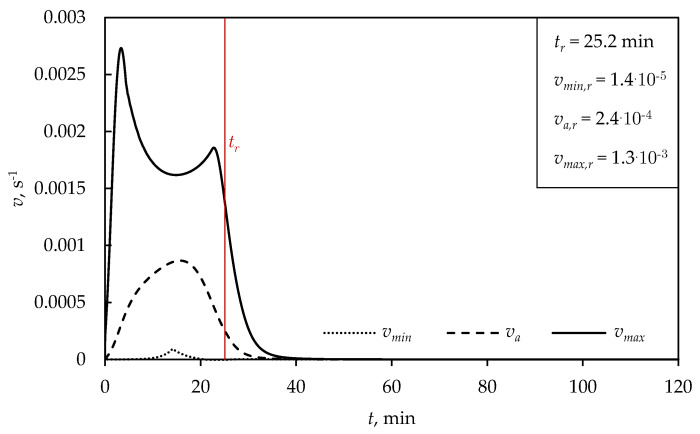
Dependence of vulcanization rate on time, for a sphere (D= 5 cm) at $T_{M}=$ 150 °C.

**Figure 5 polymers-15-01750-f005:**
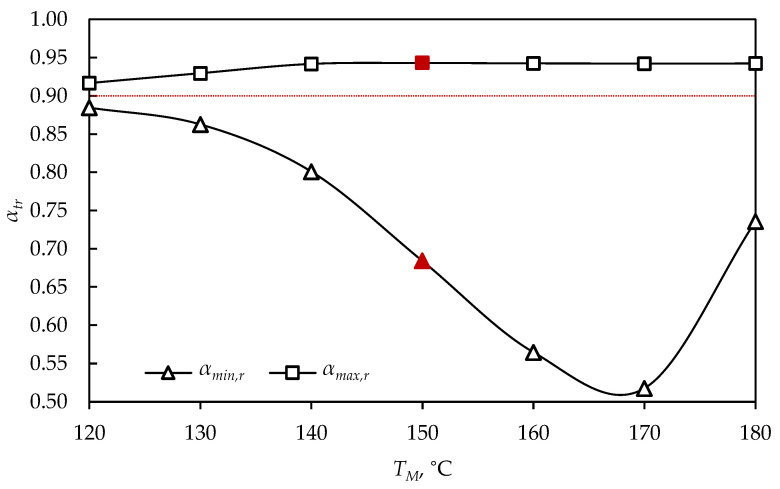
Vulcanization degree at $t_{r}$, for a sphere (*D* = 5 cm), obtained at different mold temperatures.

**Figure 6 polymers-15-01750-f006:**
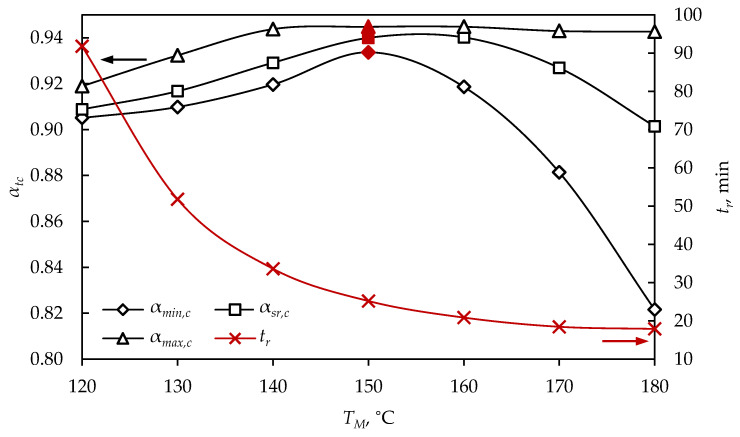
Vulcanization degree after cooling a sphere (*D* = 5 cm), obtained at different mold temperatures.

**Figure 7 polymers-15-01750-f007:**
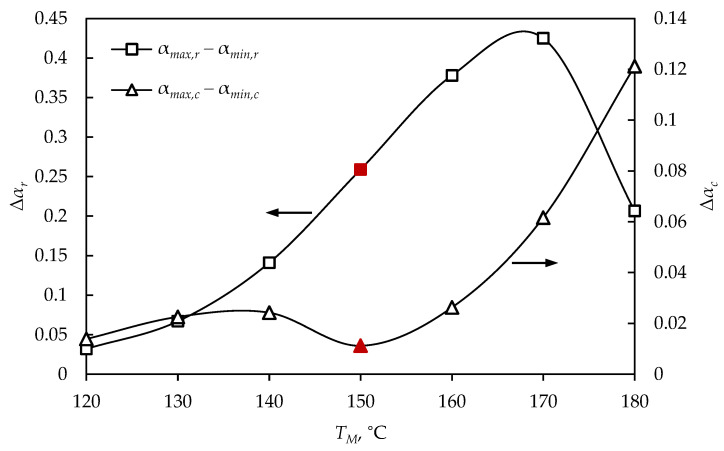
The difference between the maximal and minimal vulcanization degrees of a sphere (*D* = 5 cm) at $t_{r}$ and $t_{c}$.

**Figure 8 polymers-15-01750-f008:**
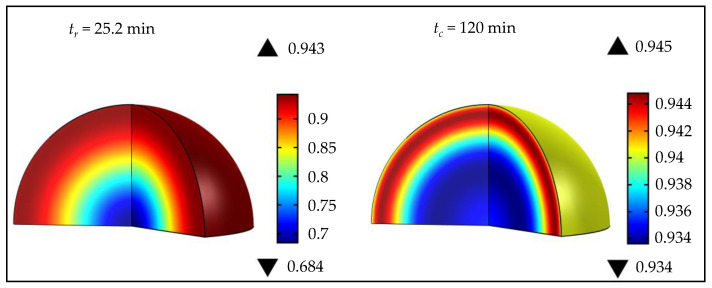
Vulcanization simulation of a sphere (D= 5 cm) at the optimal temperature (150 °C), at $t_{r}$ and $t_{c}$.

**Figure 9 polymers-15-01750-f009:**
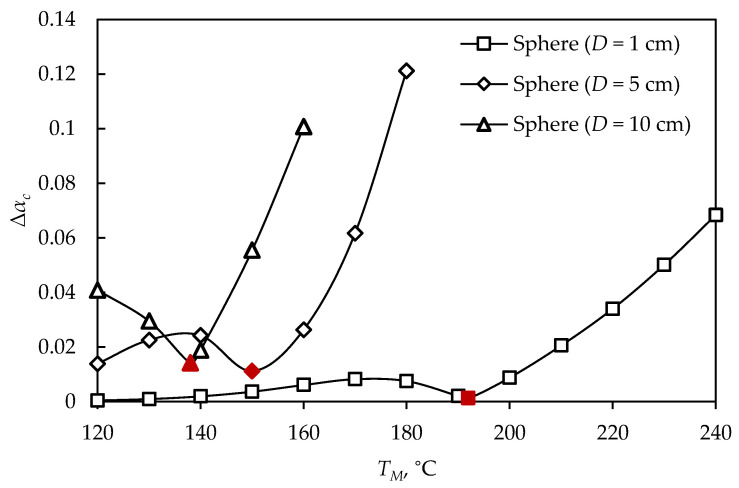
The difference between the maximal and minimal vulcanization degrees of spheres (D= 1, 5, 10 cm), at different mold temperatures.

**Figure 10 polymers-15-01750-f010:**
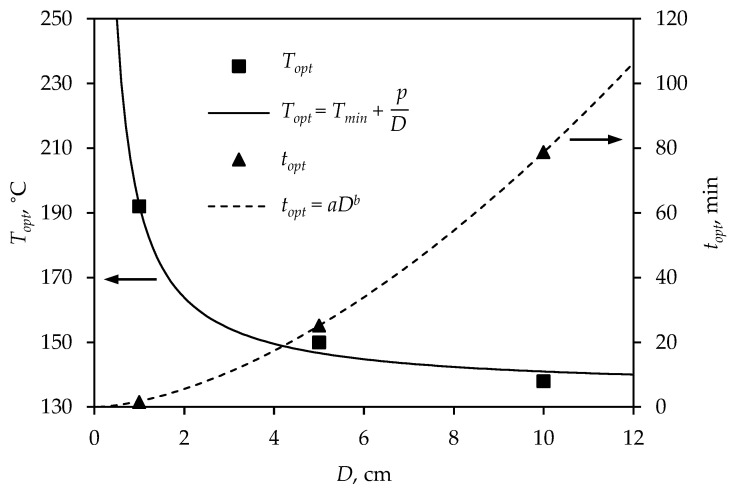
Dependence of optimal vulcanization process parameters on sphere diameter.

**Figure 11 polymers-15-01750-f011:**
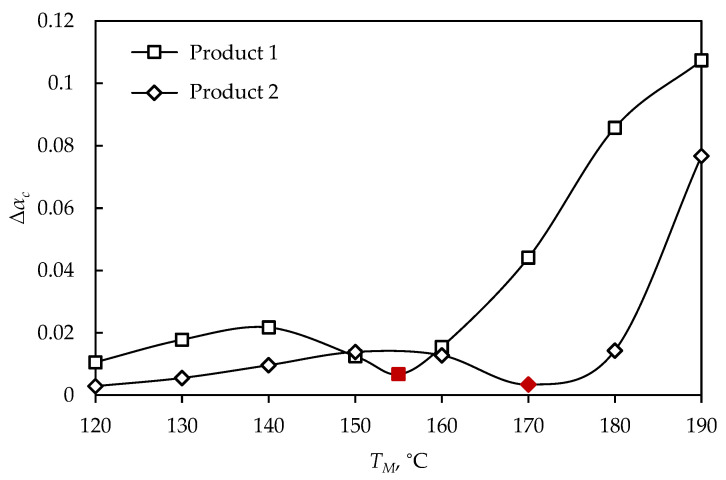
The differences between the maximal and minimal vulcanization degrees of rubber wheels at different mold temperatures.

**Figure 12 polymers-15-01750-f012:**
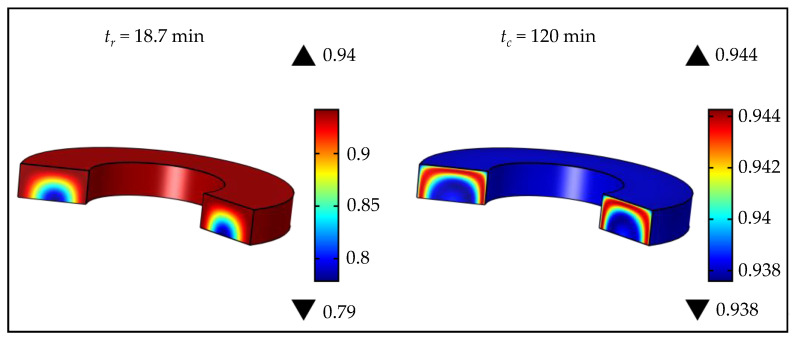
Vulcanization simulation of Product 1 at $t_{r}$ and $t_{c}$, at optimal temperature (*T_M_* = 155 °C).

**Figure 13 polymers-15-01750-f013:**
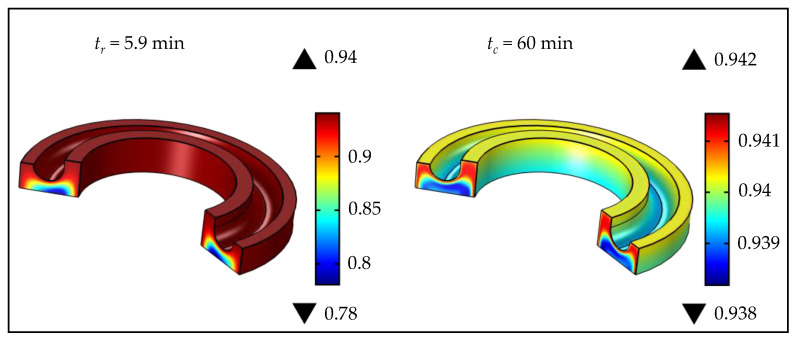
Vulcanization simulation of Product 2 at $t_{r}$ and $t_{c}$, at optimal temperature (*T_M_* = 170 °C).

**Table 1 polymers-15-01750-t001:** Coupled equations of heat transfer and kinetic model.

Heat Transfer	
Heat transfer equation	(1) $$\rho C_{p}\frac{\partial T}{\partial t}+\nabla q=Q$$
Conductivity	(2) q=−λ∇T
	(3) $$-\nabla q=\frac{1}{r}\frac{\partial }{\partial r}\left(r\lambda \frac{\partial T}{\partial r}\right)+\frac{\partial }{\partial z}\left(\lambda \frac{\partial T}{\partial z}\right)$$
Convection	(4) $$-n\cdot q=h(T_{amb}-T)$$
Heat Source	(5) $$Q=\frac{d\alpha_{c}}{dt}q_{v}\rho$$
Limit conditions and values	(6) $$h=\left\{\begin{array}{c} 10,000\;{Wm}^{-2}\;K^{-1},\alpha_{a}<0.9\\ 11.4\;{Wm}^{-2}\;K^{-1},\alpha_{a}\geq 0.9 \end{array}\right.$$
	(7) $$T_{amb}=\left\{\begin{array}{c} T_{M},\alpha_{a}<0.9\\ T_{0},\alpha_{a}\geq 0.9 \end{array}\right.$$
	$T_{0}$ = 25 °C
	$T_{M}$
	$C_{p}$ = 1.82 kJ kg${}^{-1}$ K${}^{-1}$ [38]
	ρ = 1020 kg m${}^{-3}$ [38]
	λ = 0.28 W m${}^{-1}$ K${}^{-1}$ [38]
	*K* = 0.15 mm${}^{2}$ s${}^{-1}$ [38]
	$q_{v}$ = 13 kJ kg${}^{-1}$ [40]
Solutions	*T*(*r, z, t*)
	$T_{min}\left(t\right)$, $T_{a}\left(t\right)$, $T_{max}\left(t\right)$
**Kinetic model**	
Equations	(8) $$\frac{d\alpha_{c}\left(r,z,t\right)}{dt}=A_{c}e^{-\frac{E_{ac}}{RT(r,z,t)}}\alpha_{c}{\left(r,z,t\right)}^{\frac{n(T)-1}{n(T)}}{\left(1-\alpha_{c}\left(r,z,t\right)\right)}^{\frac{n(T)+1}{n(T)}}$$
	(9) $$\frac{d\alpha_{r}\left(r,z,t\right)}{dt}=A_{r}e^{-\frac{E_{ar}}{RT(r,z,t)}}\left(x\left(T\right)-\alpha_{r}\left(r,z,t\right)\right)$$
	(10) $$\alpha \left(r,z,t\right)=\alpha_{c}\left(r,z,t\right)-\alpha_{r}\left(r,z,t\right)$$
Limit conditions and values	(11) $$n\left(r,z,t\right)=\left\{\begin{array}{c} 1,K_{n}T\left(r,z,t\right)+N_{n}\leq 1\\ K_{n}T\left(r,z,t\right)+N_{n},K_{n}T\left(r,z,t\right)+N_{n}>1 \end{array}\right.$$
	(12) $$x\left(r,z,t\right)=\left\{\begin{array}{c} 0,K_{x}T\left(r,z,t\right)+N_{x}\leq 0\\ K_{x}T\left(r,z,t\right)+N_{x},K_{x}T\left(r,z,t\right)+N_{x}>0 \end{array}\right.$$
	$N_{n}$ = −5.335 [38]
	$K_{n}$ = 0.019662 K${}^{-1}$ [38]
	$N_{x}$ = −1.479744 [38]
	$K_{x}$ = 0.003896 K${}^{-1}$ [38]
	$E_{ac}$ = 85.008 kJ mol${}^{-1}$ [38]
	$A_{c}$ = 1.017 × 10${}^{8}$ [38]
	$E_{ar}$ = 69.187 kJ mol${}^{-1}$ [38]
	$A_{r}$ = 8.639 × 10${}^{4}$ [38]
Solutions	α(r,z,t)
	$\alpha_{min}\left(t\right)$, $\alpha_{sr}\left(t\right)$, $\alpha_{max}\left(t\right)$
	$v_{min}\left(t\right)$, $v_{sr}\left(t\right)$, $v_{max}\left(t\right)$

**Table 2 polymers-15-01750-t002:** Fitting parameters in Equations (Equation 11) and (Equation 12), and quality of fitting.

	*T_min_*, °C	*p*, cm °C	*R^2^*	*MAPE, %*
$T_{opt}$ (Equation (Equation 12))	135.3	57.12	0.9866	1.25
	*a*, min	*b*	*R^2^*	*MAPE*, %
$t_{opt}$ (Equation (Equation 12))	1.769	1.648	0.9998	2.43

## Data Availability

All data generated or analyzed during this study are included in this published article.

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
