# Peer review of "A Novel Approach for Simulation and Optimization of Rubber Vulcanization"

_polymers, 2023, doi:10.3390/polym15071750_

Round 1

Reviewer 1 Report

The authors established a model to simulate the vulcanization process of NR/SBR rubbers by solving reaction kinetics and heat transfer equations via the finite element method. The model can help find the optimal process parameters that are economical, cost-effective, and energy-efficient. The work is significant for vulcanization/curing process optimizations, and the manuscript is clearly written. As such, I recommend the publication of this work. Only a minor revision is required.

(1)   In Eq. 1, only heat transfer is considered, and convection is not involved. Is the convection crucial for the vulcanization process of NR/SBR rubbers, especially for melting NR/SBR? If the authors cannot add the convection term in Eq. 1, it is also acceptable to give a discussion.

(2)   The authors used r to represent the vulcanization rate and space positions in the equations. Differentiating them is necessary.

(3)   In Figure 10, the authors used three data for the fitting. I think more points are necessary for an effective fitting.

(4)   What is the physical meaning of p, a, and b in Eqs. 11 and 12?

Author Response

The authors established a model to simulate the vulcanization process of NR/SBR rubbers by solving reaction kinetics and heat transfer equations via the finite element method. The model can help find the optimal process parameters that are economical, cost-effective, and energy-efficient. The work is significant for vulcanization/curing process optimizations, and the manuscript is clearly written. As such, I recommend the publication of this work. Only a minor revision is required.

Point 1: In Eq. 1, only heat transfer is considered, and convection is not involved. Is the convection crucial for the vulcanization process of NR/SBR rubbers, especially for melting NR/SBR? If the authors cannot add the convection term in Eq. 1, it is also acceptable to give a discussion.

Response 1: The authors would kindly like to thank for the nice comment and suggestion to improve the quality of Manuscript. Accordingly, we corrected Table 1, as well as heat transfer equation and equations related to convection - heat flux across boundaries. The following corrections are marked in Manuscript with red.

  • Heat transfer equation: Equation (1).
  • Conductivity: Equations (2) and (3).
  • Convection - heat flux across boundaries: Equation (4).
  • Heat Source: Equation (5).

Point 2: The authors used r to represent the vulcanization rate and space positions in the equations. Differentiating them is necessary.

Response 2: We would like to thank Reviewer 1 for his kind comment. According to kind comment of Reviewer 1, we defined the vulcanization rate with v and space position remained marked with r. Related corrections are marked with red in Manuscript.

Point 3: In Figure 10, the authors used three data for the fitting. I think more points are necessary for an effective fitting.

Response 3: We would like to that Reviewer 1 for very useful comment. Namely, we agree that three data points are insufficient for quality fitting. However, the proposed equations followed the theoretical constrains (D→0, Topt→∞, topt = 0 and D→∞, Topt = Tmin, topt →∞) which made this fitting more accurate. These constrains and boundary conditions are added to Manuscript (lines 375-380), as following: “Where p, a and b are the fitting parameters, and with Equations 13 and 14 the constrains and boundary conditions are set. When the diameter of the sphere tends to zero (D → 0), then the sphere becomes a point and can be exposed to an infinitely high temperature (Topt →∞), for an infinitely short time (topt = 0). On the other hand, if the sphere diameter is infinitely large (D →∞), then the sphere can be exposed to minimal temperature (Topt = Tmin) for infinitely large time (topt →∞).”

Point 4: What is the physical meaning of p, a, and b in Eqs. 11 and 12?

Response 4: We would like to thank Reviewer 1 for the comment. The obtained parameters are specific for the tested rubber mixture and can be applied to the spherical geometry. The parameters do not have actual physical meaning. However, the model presented adhered to the theoretical constraints and boundary conditions mentioned earlier, thus lending it theoretical and physical support. The constrains and boundary conditions are now added to improved Manuscript, as presented in Response 3.

Reviewer 2 Report

Please explain the innovation of this study compared with other simulation studies on vulcanization.

Author Response

Point 1: I Please explain the innovation of this study compared with other simulation studies on vulcanization.

Response 1: The authors would kindly like to thank for the nice comment and suggestion to improve the quality of Manuscript. According to suggestion, the part related to the explanation of the study innovation is added in Introduction part and significantly improved revised Manuscript.

The research paper presents several novel contributions in the field of simulation studies on rubber vulcanization:

  • The authors utilized a new model for rubber vulcanization that takes into account the phenomenon of reversion. This is a novel approach as most existing models do not consider reversion, which can lead to inaccurate predictions of the final properties of the rubber product.
  • The development of a completely new optimization procedure that takes into account cooling after removing the product from the mold is a significant novelty since it addresses a critical factor that has been overlooked in the previous optimization methods, which typically assume that the product cools instantaneously after removal from the mold. The cooling process can significantly affect the final product's properties, and by considering it in the optimization process, researchers can ensure that the final product meets desired specifications.
  • Cost-effectiveness and energy savings can be achieved by using the new model and optimization procedure. By accurately predicting properties of the final product, the new approach can reduce the need for expensive and time-consuming trial-and-error testing, which can ultimately save costs. Additionally, the new optimization procedure can reduce the energy consumption required for the vulcanization by optimizing the process parameters.
  • Another important contribution of the research is the ability to obtain homogeneous rubber products. The new model and optimization procedure can help to ensure that the final product has consistent properties throughout, which is crucial for the applications where the uniformity is important, such as for automotive tires.

Overall, the research paper contains several novel contributions that can improve the efficiency and effectiveness of the rubber vulcanization process, while also leading to cost savings and more homogeneous products.

According to Rewiever 2 comment, the following sentences are added to Introduction part (lines 80-85): “The developed optimization procedure that takes into account cooling after removing the product from the mold addresses a critical factor that has been overlooked in the previous optimization methods, which typically assume that the product cools instantaneously after removal from the mold. The novel optimization procedure can reduce the energy consumption required for the vulcanization by optimizing the process parameters.”

Reviewer 3 Report

Overall, the manuscript is well-writen and is interesting in terms of applied science/engineering of vulcanization process. However, it has some flaws commented below:

1) Introduction. line 48: Three methods can be used for vulcanization modeling and simulation the mentioned methods are actually more like approaches (i.e. how to solve the given equations). I believe authors should bring more light on different methods in terms of systems of equations used in various works. This is what makes the big difference. At least, a little qualitative comparison of kinetic models is required for the reader's best. Now it looks like if the authors are the first who propose the coupled kinetic + heat transfer equations, which is not the case.

2) The model. Why authors choose density, Cp, thermal conductivity coefficient and thermal diffusivity coefficient independent on vulcanization degree? Could you estimate an error resulting from this approximation?

3) Technical simulation parameters and quality. lines 188-191: the authors state that the mesh resolution was "particularly fine", and that it "enables precise simulation". However, the criteria of the precision is almost completely missing. I guess the choice and testing of the mesh precision should be covered with more care and objectivity, rather than a selection of knobs in the third party software. In partucular, the authors should suggest the criteria for simulations precision and demonstrate the solution reached these criteria.

4) Suggestion on additional analysis. In addition to Figs 3--9 based on minimum and maximum vulcanization degree over the volume of the sample, it's interesting to see the dependency of the volume of the undercured area (i'e' the volume with vulcanization degree below the chosen threshold value, say, 0.9) over time or over optimization parameters. I understand the minimum vulcanization degree is important, but it's also important what volume it covers, or what are vulcanization degree percentiles.

5) Software description. lines 77-84: I believe COMSOL is well-known software, and there is no need to describe it as much as in 7 lines. Please shorten the description. Please integrate information about COMSOL usage details all in one place (lines 77-84, 97-101).

After the corrections, the manuscript can be accepted to Polymers journal or to a more Applied Sci oriented journal.

Author Response

Overall, the manuscript is well-writen and is interesting in terms of applied science/engineering of vulcanization process. However, it has some flaws commented below:

Point 1: Introduction. line 48: Three methods can be used for vulcanization modeling and simulation the mentioned methods are actually more like approaches (i.e. how to solve the given equations). I believe authors should bring more light on different methods in terms of systems of equations used in various works. This is what makes the big difference. At least, a little qualitative comparison of kinetic models is required for the reader's best. Now it looks like if the authors are the first who propose the coupled kinetic + heat transfer equations, which is not the case.

Response 1: The authors would kindly like to thank for useful comment and suggestion to improve the quality of Manuscript. Accordingly, the Introduction part is now improved after providing more light on different methods in terms of systems of equations used in various works, as following (lines 46-55): “Numerous different approaches can be found in the literature for rubber kinetics studies [8-15]. Empirical, mechanistic and semi-mechanistic models represent novel kinetic models that take into account reversion phenomena. Milani and Milani [16] presented empirical model based on parabolic and hyperbolic equations, and the model's main disadvantage is lack of parameters' physical meaning. Ding and Leonov mechanistic model [17] includes reversion and enable the complete vulcanization modelling, while it is consisted of a complex reactions with more than six rate constants, which can lead to complex model to solve. Simplified reaction mechanism enables reduced number of kinetic parameters, proposed by Han [18] and Milani [19,20] in semi-mechanistic models.”

Line 61 is now corrected as: “Three approaches can be used for vulcanization modeling and simulation: analytical method, finite difference method (FDM) and finite element method (FEM).”

Additionally, following sentences are added to the Introduction (lines 80-75): “The developed optimization procedure that takes into account cooling after removing the product from the mold addresses a critical factor that has been overlooked in the previous optimization methods, which typically assume that the product cools instantaneously after removal from the mold. The novel optimization procedure can reduce the energy consumption required for the vulcanization by optimizing the process parameters.”

Point 2: The model. Why authors choose density, Cp, thermal conductivity coefficient and thermal diffusivity coefficient independent on vulcanization degree? Could you estimate an error resulting from this approximation?

Response 2: We would like to thank Reviewer 3 for his kind comment. The authors are aware that the density, Cp and thermal conductivity coefficient are dependent on temperature and vulcanization degree. Moreover, authors tested these dependences by incorporating them into the equation system for one experiment. The well-known and published dependances were utilized. The obtained results did not have significant influence on the simulation results. On the other hand, the insatiably and the divergence of the solver increased dramatically, hence, this approach was avoided. This approximation can be also found in several previously published papers by other researchers (Ghoreishy, M. H. R. A state-of-the-art review on the mathematical modeling and computer simulation of rubber vulcanization process. Iran. Polym. J. 2016, 25, 89-109., and Da Costa, H. M.; Visconte, L. L. Y.; Nunes, R. C. R.; Furtado, C. R. G. Rice husk ash filled natural rubber. I. Overall rate constant determination for the vulcanization process from rheometric data. J. Appl. Polym. Sci. 2003, 87, 1194-1203.)

The following part is added to the manuscript (lines 93-95): “Based on literature review [5,35], density, thermal conductivity coefficient and thermal diffusivity coefficient were taken as a constant, without causing the significant error of the simulation results.”

Point 3: Technical simulation parameters and quality. lines 188-191: the authors state that the mesh resolution was "particularly fine", and that it "enables precise simulation". However, the criteria of the precision is almost completely missing. I guess the choice and testing of the mesh precision should be covered with more care and objectivity, rather than a selection of knobs in the third party software. In partucular, the authors should suggest the criteria for simulations precision and demonstrate the solution reached these criteria.

Response 3: The authors would kindly like to thank for the nice comment and suggestion to improve the quality of Manuscript. Using a less refined mesh resulted in a lower resolution solution, whereas increasing the level of detail in the mesh significantly increased simulation time, but did not provide significant accuracy improvements. The solver convergence criteria and tolerance for simulations were set in COMSOL. The precision of simulation and mesh can be only tested under real production conditions by experiments that include comprehensive study of rubber product cross-section, in order to evaluate vulcanization degree and temperature profiles for different molding times. The authors are currently working on developing the approach that will be the part of future research.

According to previous discussion, the adequate explanation is added to revised Manuscript (lines 204-205), as following: “In this work, it was used "Physics Controlled Mash" with element size set to "Extra fine", defined in the software COMSOL Multiphysics, which enables precise simulation, and, as the tested products are symmetric along a fixed axis, this type of geometric mesh does not lead to long simulation time when used only in the domain shown in Figure 1.”

Point 4: Suggestion on additional analysis. In addition to Figs 3--9 based on minimum and maximum vulcanization degree over the volume of the sample, it's interesting to see the dependency of the volume of the undercured area (i'e' the volume with vulcanization degree below the chosen threshold value, say, 0.9) over time or over optimization parameters. I understand the minimum vulcanization degree is important, but it's also important what volume it covers, or what are vulcanization degree percentiles.

Response 4: Thank You very much for the great idea and suggestion. The authors decided to use the criteria of average curing degree of 0.9 in order to achieve the stability of rubber product during the removal from the mold. This approach ensures that sufficient product volume is above the vulcanization degree threshold of 0.9. As a result, using the range between the minimum and maximum vulcanization degree can be an appropriate method for assessing the product uniformity. However, we will also consider the volume fraction as other criteria for the future optimisation regarding more complex geometries.

According to previous discussion, more detailed explanation is added to revised Manuscript (lines 122-127): “The constrain for removing the product is reaching an average vulcanization degree of 0.9, in order to achieve the stability of rubber product during the removal from the mold. This approach ensures that sufficient product volume is above the vulcanization degree threshold of 0.9. The ambient temperature (Tamb) was defined by Equation 7, according to the average vulcanization degree of 0.9, in a way to ensure a product with a defined shape.”

Point 5: Software description. lines 77-84: I believe COMSOL is well-known software, and there is no need to describe it as much as in 7 lines. Please shorten the description. Please integrate information about COMSOL usage details all in one place (lines 77-84, 97-101).

Response 5: We would like to thank Reviewer 3 for his useful suggestion. Accordingly, we shorten description related to COMSOL, and integrated information about COMSOL usage details all in one place in revised Manuscript.

Reviewer 4 Report

The manuscript submitted by Lubura et al. describes a method to simulate and optimize the vulcanization of NR/SBR based on a previously obtained formulation. The work is interesting, and the results are definitively suitable for readers of Polymers. However, to increase the impact of the work, some suggestions are provided herein:

1) The NR/SBR literature is very extensive, and the introduction does not provide significant referencing. Several arguments in the text lack a reference in the current version. Please, amend this in the revised version.

2) Line 22, how high is the energy consumption during vulcanization? Is this step a sustainability drawback of the rubber industry?

3) The novelty of the approach versus other works is still unclear. 

4) Temperature is described as one key component of rubber vulcanization. What about crosslinkers, activators, accelerators, and rubber types?

5) Regarding point 4, how reproducible are the results of the models presented considering different formulations?

6) More details about the recipes used should be described in the materials.

7) How does the modeling data correlates to a regular vulcanization curve from a vulcameter?

8) Line 264, what are the industrial standards referred to?

9) Lines 439, the optimization approach was applied to a product shape, but were the properties tested and compared experimentally? 

Author Response

The manuscript submitted by Lubura et al. describes a method to simulate and optimize the vulcanization of NR/SBR based on a previously obtained formulation. The work is interesting, and the results are definitively suitable for readers of Polymers. However, to increase the impact of the work, some suggestions are provided herein:

Point 1: The NR/SBR literature is very extensive, and the introduction does not provide significant referencing. Several arguments in the text lack a reference in the current version. Please, amend this in the revised version.

Response 1: We would like to thank Reviewer 4 for his kind observation. According to the comment, the authors extended Introduction of revised Manuscript (lines 17-19_ by adding more literature data on NR/SBR systems, as following: “Natural rubber and styrene-butadiene rubber blend has been extensively used for optimizing the compound properties with the required mechanical-dynamical properties of final products [1,2].”

Point 2: Line 22, how high is the energy consumption during vulcanization? Is this step a sustainability drawback of the rubber industry?

Response 2: We would like to thank Reviewer 4 for his kind observation. The vulcanization process and molding include heating for predefined period of time that lead to energy consumption. During the mass production of rubber goods, any slight change in temperature and time can lead to significant energy savings. According to the comment of Reviewer and related discussion, authors emphasized the need for rubber production optimization in revised Manuscript, where the sentence in Line 22 is now corrected and expanded: “The vulcanization process and molding include heating for predefined period of time that lead to energy consumption. Hereby, optimization of parameters of energy demanding vulcanization process is of crucial importance, in order to achieve economical process management and to obtain high quality rubber products.”

Point 3: The novelty of the approach versus other works is still unclear.

Response 3: We would like to thank Reviewer 4 for his kind comment. Revised Manuscript is now improved by emphasizing the novelty of the approach versus other works in Introduction part (lines 46-55): “Numerous different approaches can be found in the literature for rubber kinetics studies that include reversion phenomena [8-15]. Empirical, mechanistic and semi-mechanistic models represent novel kinetic models that take into account reversion phenomenon. Milani and Milani [16] presented empirical model based on parabolic and hyperbolic equations, and the model's main disadvantage is lack of parameters' physical meaning. Ding and Leonov mechanistic model [17] includes reversion and enable the complete vulcanization modelling, while it is consisted of a complex reactions with more than six rate constants, which can lead to complex model to solve. Simplified reaction mechanism enables reduced number of kinetic parameters, proposed by Han [18] and Milani [19,20] in semi-mechanistic models.”

Additionally, following sentences are added to the Introduction (lines 80-75): “The developed optimization procedure that takes into account cooling after removing the product from the mold addresses a critical factor that has been overlooked in the previous optimization methods, which typically assume that the product cools instantaneously after removal from the mold. The novel optimization procedure can reduce the energy consumption required for the vulcanization by optimizing the process parameters.”

Point 4: Temperature is described as one key component of rubber vulcanization. What about crosslinkers, activators, accelerators, and rubber types?

Response 4: We would like to thank Reviewer 4 for his kind comment. The goal of this approach was to create the general procedure for modeling, simulation and optimization that can be applied on all vulcanization systems and curing curves. Hence, in this work, the authors were focused on the study of temperature and time influence on the vulcanization optimization, having in mind that the rubber type and composition has, also, the certain influence on the vulcanization, due to different kinetic parameters derived from fitting the vulcameter curves with new mathematical model. Once the kinetic parameters were obtained, the proposed simulation and optimization can be conducted for any vulcanization system. The model is describing all phenomena during the vulcanization and calculates the kinetic parameters, accordingly. The obtained parameters are incorporated into the model and solved together with heat transfer equations. Finally, the results are used for estimation of optimal vulcanization temperature and time for specific products.

Point 5: Regarding point 4, how reproducible are the results of the models presented considering different formulations?

Response 5: We would like to thank Reviewer 4 for his nice comment. The mathematical model was sucefully and independently verified on different formulations: Lopes, H., Silva, S.P., Carvalho, J.P. and Machado, J., 2022. A new modelling approach for predicting process evolution of cork-rubber composites slabs vulcanization. Scientific Reports, 12(1), p.8002.

Point 6: More details about the recipes used should be described in the materials.

Response 6: We would like to thank Reviewer 4 for his useful comment. The authors added more information in Experimental part of Manuscript, as following: “The properties (specific heat, density, thermal conductivity coefficient and thermal diffusivity coefficient) of a commercially available mixture of natural and styrene-butadiene rubber, based on a conventional vulcanization system with with sulfur as curing agent, N-cyclohexyl-2-benzothiazyl-sulfenamide (CBS) as an accelerator, zinc oxide and stearic acid as curing activators, and 40 phr of and carbon black as a filler, were assessed in the previous research, and presented in Table 1.”

Point 7: How does the modeling data correlates to a regular vulcanization curve from a vulcameter?

Response 7: We would like to thank Reviewer 4 for his useful comment. The model fitting was performed on data obtained from vulcameter with great success. Detailed information on model testing and derivation can be found in the following paper: Bera, O., Pavličević, J., Ikonić, B., Lubura, J., Govedarica, D. and Kojić, P., 2021. A new approach for kinetic modeling and optimization of rubber molding. Polymer Engineering & Science, 61(3), pp.879-890.

Regular vulcanization curves (without the reversion) can be also described using the new model. The difference will be in the reversion part of the model that can be excluded. Moreover, without the exclusion of reversion part, the fitting will result in the negligible values for reversion rate constant, confirming the low reversion.

Point 8: Line 264, what are the industrial standards referred to?

Response 8: The authors would like to thank to Reviewer 4 for his useful comment. Accordingly, revised Manuscript is now corrected, as following (lines 278-281): “The values of the vulcanization degree after cooling the sphere (D = 5 cm) are above 0.93, and the vulcanization degree is high throughout the interior of the sphere, satisfying the set criteria of 0.9.”

Point 9: Lines 439, the optimization approach was applied to a product shape, but were the properties tested and compared experimentally?

Response 9: The authors would kindly like to thank for the nice comment and suggestion to improve the quality of Manuscript. The optimization results (vulcanization temperature and time) for product shapes were tested and confirmed experimentally during the rubber production. The proposed parameters resulted in the rubber products with desired properties (hardness, abrasion, shape consistence), while the implementation of optimal parameters reduced the energy consumption. Further research will consist of experiments that includes comprehensive study of rubber product cross-section in order to evaluate vulcanization degree and temperature profiles at different molding times. According to the previous discussion, the last sentence in Conclusion part is now changed: “The assessed optimization results (vulcanization temperature and time) for investigated product shapes were successfully tested and confirmed experimentally during the rubber production, allowing the application of the proposed optimization approach for obtaining vulcanization parameters of various vulcanization systems and rubber products with complex shapes. The proposed parameters resulted in the rubber products with desired properties, while the implementation of optimal parameters in the real industrial surrounding enables reduction of the energy consumption.”

Round 2

Reviewer 3 Report

Points 1, 4 and 5:
Thanks for the answer.

Point 2/Response 2 -- Answer:
I would like to ask the authors to add the Supporting Information to the manuscript, showing the tests of dependencies authors claimed. I have no mistrust on these results, but I believe this type of negative results are also of great value for the reader and is important and crucial part of research that should be demonstrated. Please include the expressions for the tested dependencies, the short description of the test task, the metrics used to quantify the difference with constant approximation, and the results.
I understand that the similar data is already shown in different papers, but you claim a better model, so it's important to demonstrate it for this model as well.

Point 3/Response 3 -- Answer:
About the precision of simulations, there are two different precision metrics that could be applied to the simulations data: the presicion of the numerical solution and the precision of the model itself. The authors discuss the precision of the model in their answer, which I agree is hard to evaluate, as it requires a very specific experimental data. However, initially I asked about the solution precision, which can and must be quantified solely by comparing numerical solutions. I still want to encourage the authors to clarify the meanings of terms in lines 196--208. "enables precise simulation" - how the authors define 'precise' for themselves? Is it just a knob in COMSOL, and if COMSOL says precise, it is just directly considered precise? Or there are some analythical evidence of the 'precise' simulations to be more precise? In the same way, "does not lead to long simulation time" - what is 'long simulation time for the authors? Please make it more concrete, write the wall clock time of you simulations and the resources used, to make it understandable for the future reader.
Also please check the misprint "Physics Controlled Mash" (should be Mesh).

Author Response

Points 1, 4 and 5:

Thanks for the answer.

Point 2/Response 2 -- Answer: I would like to ask the authors to add the Supporting Information to the manuscript, showing the tests of dependencies authors claimed. I have no mistrust on these results, but I believe this type of negative results are also of great value for the reader and is important and crucial part of research that should be demonstrated. Please include the expressions for the tested dependencies, the short description of the test task, the metrics used to quantify the difference with constant approximation, and the results.

I understand that the similar data is already shown in different papers, but you claim a better model, so it's important to demonstrate it for this model as well.

Point 2/Response 2 -- Response: The authors would kindly like to thank for the nice comment and suggestion to improve the quality of Manuscript. According to suggestion, the part of the explanation is added to the Manuscript (lines 93-101): “The density, thermal conductivity coefficient, and thermal diffusivity coefficient were treated as constants based on a review of relevant literature [5,35]. This decision did not significantly affect the simulation results, as the standard deviation for the constant values of the thermal conductivity coefficient and the thermal diffusivity coefficient was only 0.0074 W m-1 K-1 and 0.0036 mm2 s-1, respectively. Attempting to incorporate an appropriate temperature and vulcanization degree dependence for these parameters resulted in a divergence in certain simulations. Therefore, the decision to use constant values was based on the low standard deviation and the avoidance of divergence.”

Point 3/Response 3 -- Answer: About the precision of simulations, there are two different precision metrics that could be applied to the simulations data: the presicion of the numerical solution and the precision of the model itself. The authors discuss the precision of the model in their answer, which I agree is hard to evaluate, as it requires a very specific experimental data. However, initially I asked about the solution precision, which can and must be quantified solely by comparing numerical solutions. I still want to encourage the authors to clarify the meanings of terms in lines 196--208. "enables precise simulation" - how the authors define 'precise' for themselves? Is it just a knob in COMSOL, and if COMSOL says precise, it is just directly considered precise? Or there are some analythical evidence of the 'precise' simulations to be more precise? In the same way, "does not lead to long simulation time" - what is 'long simulation time for the authors? Please make it more concrete, write the wall clock time of you simulations and the resources used, to make it understandable for the future reader.

Also please check the misprint "Physics Controlled Mash" (should be Mesh).

Point 3/Response 3 – Response: The authors would kindly like to thank for useful comment and suggestion to improve the quality of Manuscript. Accordingly, the Materials and Methods part is now improved after providing more light on simulation in COMSOL Multiphysics software, as following (lines 214-221): “The simulations were carried out using selected physics modules with a controlled tolerance for relative error of 0.01. The absolute tolerance was defined using the ”Scaled” Global method with a default tolerance factor of 0.1. In the ”Scaled” method, the absolute tolerance is applied to scaled variables, while the “factor tolerance” method makes the absolute tolerance proportional to the relative tolerance. The simulations can be considered precise in terms of mesh definition and error, as the solver was run using default settings, resulting in a short simulation time of approximately 10 minutes.”

The authors would like to thank Reviewer 3 for observing the spelling mistake, “Physics Controlled Mash” is now changed to “Physics Controlled Mesh”.

Reviewer 4 Report

The manuscript has been improved and responded to the issues raised by the reviewers. 

Author Response

The manuscript has been improved and responded to the issues raised by the reviewers. 

Response: We would like to thank Reviewer 4 for his kind comment.